# A glutamine sensor that directly activates TORC1

Mirai Tanigawa[1], Katsuyoshi Yamamoto[2], Satoru Nagatoishi [3], Koji Nagata [4], Daisuke Noshiro [5], Nobuo N. Noda [5], Kouhei Tsumoto[3,6] & Tatsuya Maeda [1,2✉]

TOR complex 1 (TORC1) is an evolutionarily-conserved protein kinase that controls cell growth and metabolism in response to nutrients, particularly amino acids. In mammals, several amino acid sensors have been identified that converge on the multi-layered machinery regulating Rag GTPases to trigger TORC1 activation; however, these sensors are not conserved in many other organisms including yeast. Previously, we reported that glutamine activates yeast TORC1 via a Gtr (Rag ortholog)-independent mechanism involving the vacuolar protein Pib2, although the identity of the supposed glutamine sensor and the exact TORC1 activation mechanism remain unclear. In this study, we successfully reconstituted glutamine-responsive TORC1 activation in vitro using only purified Pib2 and TORC1. In addition, we found that glutamine specifically induced a change in the folding state of Pib2. These findings indicate that Pib2 is a glutamine sensor that directly activates TORC1, providing a new model for the metabolic control of cells.

[1] Departments of Biology, Hamamatsu University School of Medicine, Shizuoka, Japan. [2] Institute for Quantitative Biosciences, The University of Tokyo, Tokyo, Japan. [3] The Institute of Medical Science, The University of Tokyo, Tokyo, Japan. [4] Department of Applied Biological Chemistry, Graduate School of Agricultural and Life Sciences, The University of Tokyo, Tokyo, Japan. [5] Institute of Microbial Chemistry (BIKAKEN), Tokyo, Japan. [6] School of Engineering, The University of Tokyo, Tokyo, Japan. ✉email: tmaeda@hama-med.ac.jp

Glutamine is the most abundant amino acid in many eukaryotic cells[1–3], which supplies an amino group for the biosynthesis of various metabolites including other amino acids and nucleotides. In addition, glutamine is also a respiratory fuel that acts as a precursor for α-ketoglutarate (αKG) in the tricarboxylic acid cycle. Thus, it has been proposed that cells monitor intracellular glutamine levels as an indicator of cellular nitrogen status and adjust their metabolism accordingly to maintain cellular homeostasis[4,5]. Indeed, glutamine has been shown to regulate cellular metabolism at least partly by activating the target of rapamycin complex 1 (TORC1), which is a central regulator of cell growth and metabolism[6–8].

TORC1 is activated in response to various nutritional cues, including amino acids, and activated TORC1 promotes cell growth by activating anabolic processes such as protein synthesis and inhibiting catabolic processes such as autophagy[9,10]. However, the mechanism via which amino acids induce TORC1 activation is poorly understood. Recent studies have shown that amino acids activate TORC1 via plural-independent pathways[11], among which the most well-studied pathway involves evolutionarily-conserved heterodimeric small GTPases: RagA/B-RagC/D (the Rag complex) in mammals and Gtr1-Gtr2 (the Gtr complex) in yeast[12–14]. Sestrin2 and CASTOR were recently identified as leucine and arginine sensors, respectively, that trigger Rag complex activation in mammals[15,16] but are unlikely to be conserved in other organisms such as yeast.

Alongside other groups, we recently revealed that glutamine activates yeast TORC1 via a Gtr complex-independent mechanism involving the vacuolar protein Pib2[17–20], which resides on the vacuolar membrane by virtue of its phosphatidylinositol 3-phosphate (PI(3)P)-interacting FYVE domain[17]. Along with the FYVE domain, conserved "E" and "tail" motifs of unknown function in Pib2 have been reported to be necessary for glutamine-responsive TORC1 activation[17,19,20]; however, the precise role of Pib2 in TORC1 activation remains unknown.

Although neither the Gtr complex nor Pib2 is essential for yeast cell growth, their double knockout results in synthetic lethality that is fully rescued by hyperactive mutations in TOR1, which encodes a catalytic component of TORC1[17]. Thus, these observations suggest that yeast TORC1 is activated mainly via pathways involving either the Gtr complex or Pib2, which are referred to hereafter as the Gtr and Pib2 pathways, respectively.

Unlike mammalian/mechanistic TORC1 (mTORC1), whose cellular localization changes in response to amino acid availability[13], yeast TORC1 constitutively resides on the vacuolar membrane[21]. Indeed, we previously developed an in vitro kinase assay to exclusively monitor Pib2 pathway-mediated TORC1 activation using purified vacuoles as a TORC1 source, which displayed glutamine-responsive TORC1 activation[18]. These observations indicate that the vacuole contains all the essential components of the Pib2 pathway, including the as-yet-unidentified glutamine sensor. In addition, they also suggest that the glutamine sensor monitors not extracellular but intracellular glutamine levels. Since intracellular glutamine levels are exceedingly high (up to 15–35 mM)[3,22,23], the affinity between the putative sensor and glutamine must be extremely low to appropriately monitor and respond to physiological changes in intracellular glutamine levels. However, we showed that Pib2-dependent TORC1 activation is highly glutamine-specific[18], which is seemingly contradictory as higher protein-ligand interaction specificity generally results in higher interaction affinity. Thus, it remains unclear how the unidentified glutamine sensor reconciles high specificity with low affinity. Since some metabolites including ATP and NADH also exist at an mM intracellular concentration, elucidating the glutamine-sensing mechanism in

the Pib2 pathway would provide a model for the general sensing mechanism of high concentration metabolites.

In this study, we show that Pib2 alone is sufficient for the glutamine-responsive activation of TORC1 in vitro. Our findings indicate that Pib2 plays dual roles as a glutamine sensor and a direct activator of TORC1, representing an unprecedented amino acid–responsive activation mechanism of TORC1.

## Results

**In vitro reconstitution of the glutamine-responsive Pib2-TORC1 interaction.** Previously, we reported that Pib2 and TORC1 physically interact in the presence of glutamine[18]. To further characterize this interaction, we performed in vitro pull-down assays with bacterially expressed and purified GST-tagged Pib2 (GST-Pib2). The C-terminal half of Pib2 (304-635) (Fig. 1a) was used as truncation improved the solubility and proteolytic-lability of the recombinant Pib2 protein while retaining glutamine-responsiveness[19,20]. When TORC1 in yeast cell lysates was pulled-down with GST-Pib2 without any amino acids, very little TORC1 co-precipitated with GST-Pib2 (Fig. 1b); however, adding L-glutamine to the cell lysate enhanced this interaction, as did L-cysteine to a lesser extent but not D-glutamine, L-asparagine, nor L-leucine (Fig. 1b). This observation is consistent with our previous finding that only L-glutamine and L-cysteine activate TORC1 in vitro in a Pib2-dependent manner[18].

Next, we examined the dose-dependent effects of glutamine on the Pib2-TORC1 interaction (Fig. 1c) and on TORC1 activation (Fig. 1d). To measure TORC1 activity, we used an in vitro kinase assay using permeabilized yeast cells as the TORC1 kinase source[18]. The assay monitors only Pib2-dependent TORC1 activation since permeabilized cells prepared from pib2Δ cells do not respond to glutamine[18]. Both the Pib2-TORC1 interaction (Fig. 1c) and TORC1 activation (Fig. 1d) displayed similar dose responses to glutamine, with a concentration over 10 mM required to fully induce responses, comparable to intracellular levels[3,22,23]. Glutamine dose-responsive Pib2-TORC1 interaction has also been reported previously[20]. Together, these observations support the hypothesis that the Pib2-TORC1 interaction induced by glutamine triggers TORC1 activation[18,20].

The results of the in vitro pull-down assays (Fig. 1d) indicated that vacuolar membranes are not necessary for the glutamine-induced Pib2-TORC1 interaction if sufficient Pib2 is supplied, suggesting that no essential components other than Pib2 and TORC1 are required for the glutamine-responsive interaction. To test this hypothesis, we performed pull-down assays in which yeast lysates were replaced with TORC1 purified using TAP-tagged Kog1, which is the yeast orthologue of Raptor (Fig. 1e)[24]. Glutamine addition enhanced the Pib2-TORC1 interaction (Fig. 1f, g), indicating that only Pib2 and TORC1 are required for their glutamine-induced interaction and suggesting that either or both of Pib2 and TORC1 act as the glutamine sensor(s).

**The Pib2 E motif is required for the glutamine-induced Pib2 interaction with TORC1.** To explore the regions on Pib2 that are required for the glutamine-induced Pib2-TORC1 interaction, we performed in vitro pull-down assays with a series of Pib2 deletion mutants (Fig. 2a, b). Pib2(304-620), which lacks the C-terminal tail motif essential for TORC1 activation[17,19,20], still interacted with TORC1 in response to glutamine (Fig. 2b), indicating that the tail motif is dispensable for the glutamine-responsiveness of the Pib2-TORC1 interaction. In contrast, Pib2(440-635), which lacks the E motif required for both the TORC1 interaction and activation, did not respond to glutamine (Fig. 2b), suggesting that the E motif is necessary for the glutamine-induced Pib2-TORC1

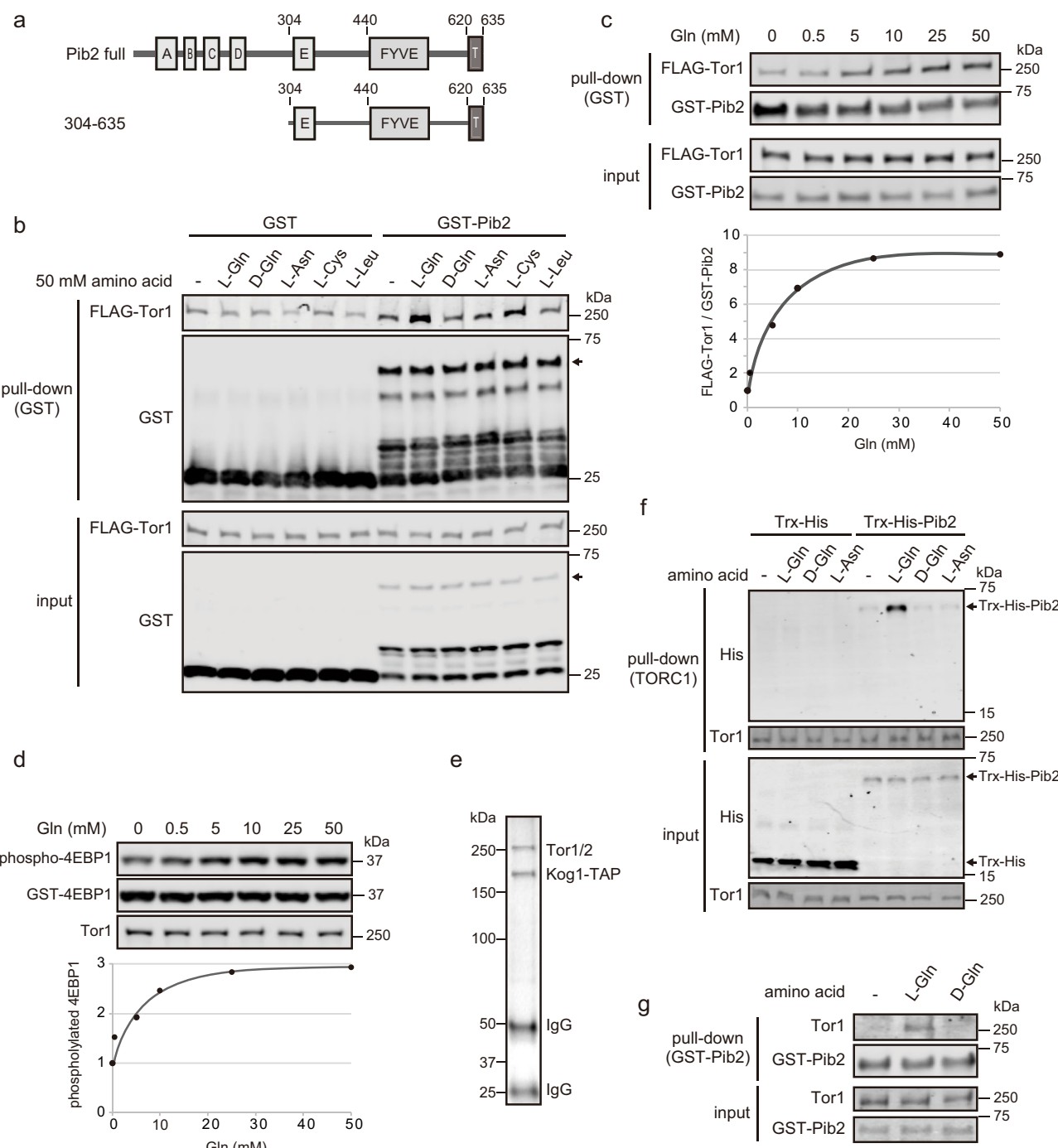

**Fig. 1 In vitro reconstitution of the glutamine-responsive Pib2-TORC1 interaction. a** Schematic diagram of full-length Pib2 with defined motifs and the truncation mutant used in (**b**, **c**, **f**, and **g**). **b** L-glutamine and L-cysteine enhance the Pib2-TORC1 interaction in vitro. Lysates from FLAG-Tor1 expressing yeast cells (TS269) were subjected to pull-down assays with bacterially-expressed GST-Pib2(304-635) and the indicated amino acids (final concentration 50 mM). **c** The Pib2-TORC1 interaction depends on glutamine dose. Pull-down assays were performed as in (**b**), except that the indicated concentrations of L-glutamine were added to the cell lysates. The line graph shows GST-Pib2-bound FLAG-Tor1 normalized to sample without glutamine. **d** In vitro Pib2-dependent TORC1 activity depends on glutamine dose. In vitro TORC1 kinase assays monitored Pib2-dependent TORC1 activation using permeabilized yeast cells as the kinase source with the indicated concentrations of L-glutamine. The line graph shows the ratio of phosphorylated 4EBP1 to total 4EBP1 normalized to sample without glutamine. **e** Silver-stained SDS-PAGE gel for TORC1 purified from Kog1-TAP-expressing yeast cells (RL171-2a) using IgG-coupled magnetic beads. **f** Pib2 and TORC1 are sufficient to induce their glutamine-induced interaction. Bacterially-expressed and purified Trx-His-Pib2(304-635) and TORC1 purified from yeast on magnetic beads were incubated with the indicated amino acids (30 mM). Trx-His-Pib2(304-635) co-precipitation with TORC1 was detected by western blotting. **g** Induction of the Pib2-TORC1 interaction by glutamine is stereospecific. TORC1 was purified as in (**e**) and eluted from the beads using tobacco etch virus protease before being incubated with bacterially-expressed GST-Pib2(304-635) and the indicated amino acids (30 mM). TORC1 co-precipitation with GST-Pib2 was detected by western blotting.

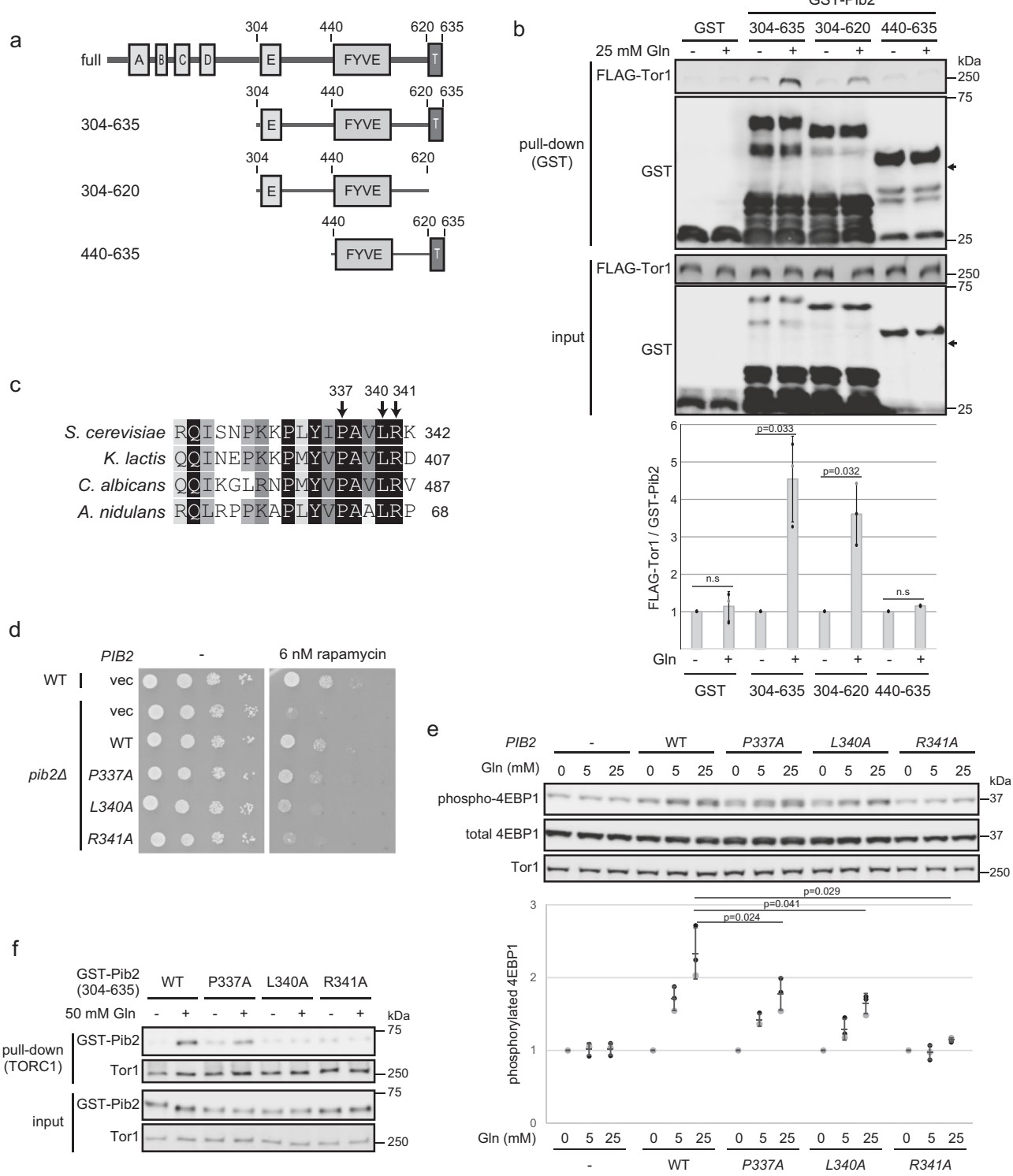

interaction. To confirm this hypothesis, we introduced point mutations into three conserved residues in the E motif (Fig. 2c) and evaluated the functionality of the mutants in vivo and in vitro. The pib2(R341A) mutant failed to complement rapamycin sensitivity in *pib2Δ* cells, whereas the *pib2(P337A)* and *pib2(L340A)* mutants partially complemented rapamycin sensitivity (Fig. 2d). Consistently, *Pib2(R341A)* displayed the most severely impaired TORC1 activation of all the mutants (Fig. 2e) and failed to interact with TORC1 even in the presence of glutamine (Fig. 2f). Together, these results indicate that the E motif

is essential for the glutamine-induced interaction between Pib2 and TORC1.

**Hyperactivating *PIB2* mutations are clustered in the conserved tail motif.** Since purified Pib2 interacts with purified TORC1 (Figs. 1f, g, and 2f), Pib2 is likely to directly activate TORC1. To obtain clues for how Pib2 activates TORC1, we screened for hyperactive *PIB2* mutants. Our initial attempts yielded N-terminal deletion mutants, consistent with previous reports in

**Fig. 2 The Pib2 E motif is required for its glutamine-induced interaction with TORC1. a** Schematic diagram of full-length Pib2 and the deletion mutants is used in (**b**). **b** The E-motif containing the central region of Pib2(304-439) is required for the glutamine-induced interaction between Pib2 and TORC1. Pull-down assays were performed using yeast cell lysates and bacterially-expressed GST-Pib2 deletion mutants (as in Fig. 1b). The bar graph shows GST-Pib2-bound FLAG-Tor1 normalized to the samples without glutamine. Error bars represent the standard deviation ($n = 3$ independent experiments), and significance was determined by a two-tailed Student's $t$-test. **c** The E motif in Pib2 is conserved among species. Sequence alignment of the E motifs in Pib2-like proteins from the indicated species. **d** Pib2 E motif mutants display rapamycin sensitivity. Wild-type (TM141) or pib2Δ (MH1059) strains were transformed with a vector (pRS416) or plasmids encoding wild-type *PIB2* (pMH342) or pib2 mutants (pMH497, pMH499, or pMH498). Serially diluted cell suspensions were spotted on SD plates lacking uracil with or without 6 nM rapamycin and grown at 30 °C for 3 days. **e** Pib2 E motif mutants decrease TORC1 activity in vitro. An in vitro TORC1 kinase assay was performed (as for Fig. 1d) with permeabilized yeast cells prepared from pib2Δ (MH1059) cells carrying a vector (p416ADH) or plasmids encoding wild-type *PIB2* (pMH330) or pib2 mutants (pMH510, pMH512, or pMH514). Dot plots represent the ratio of phosphorylated/total 4EBP1 normalized to the samples without glutamine ($n = 3$ independent experiments). **f** Pib2 E motif mutants are deficient for the glutamine-induced interaction with TORC1. Purified TORC1 on magnetic beads was incubated with bacterially-expressed GST-Pib2(304-635) mutants with or without 30 mM glutamine. GST-Pib2 co-precipitation with TORC1 was detected by western blotting.

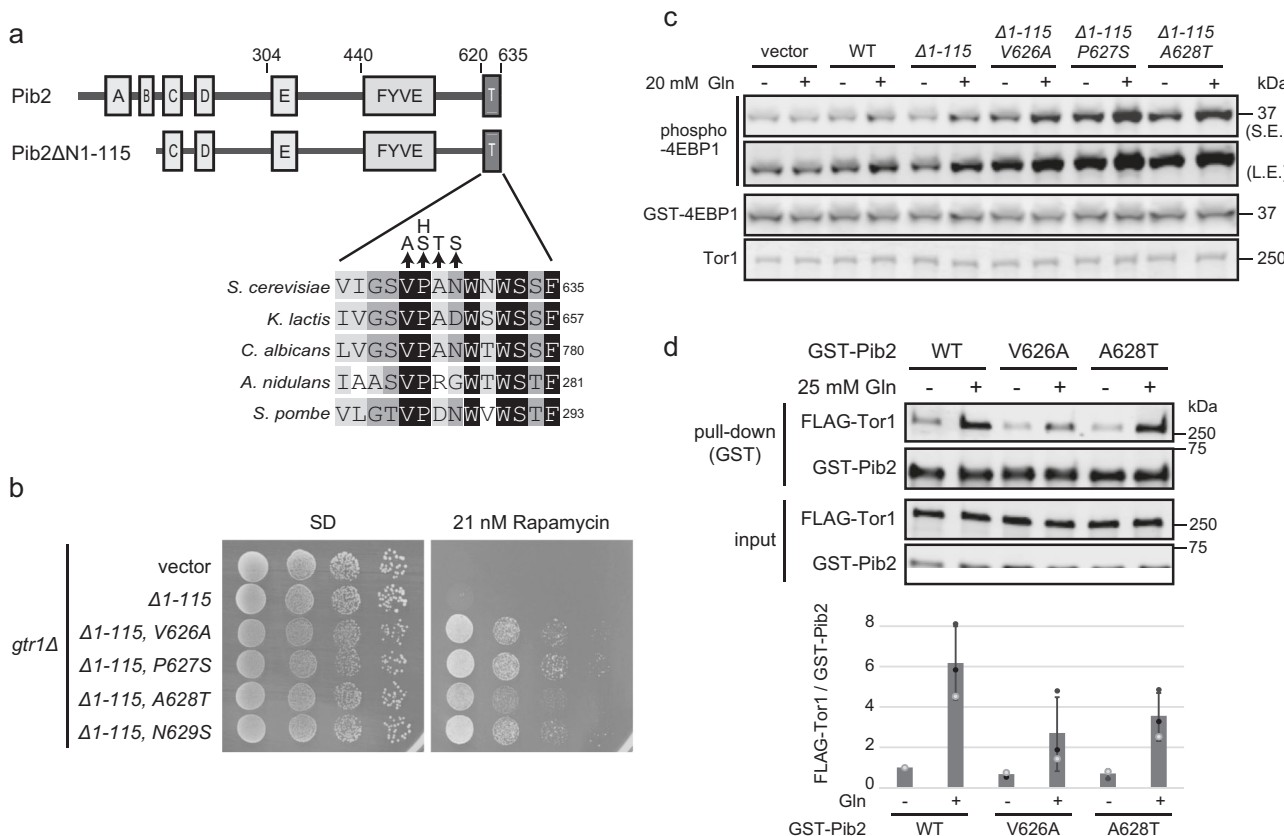

**Fig. 3 Active *PIB2* mutations are clustered in the conserved tail motif. a** Schematic diagram of wild-type Pib2 and Pib2(Δ1−115) was used for active mutant screening. Mutations in the active mutants from the tail motif are shown with multiple sequence alignments for different species. **b** Tail motif mutations confer yeast cells with rapamycin resistance. The gtr1Δ strain (MH1030) was transformed with a vector (p416ADH) or plasmids encoding *PIB2* mutants (pMH334, pKY11-1, pKY14-1, pKY24-1, or pKY21). Serially diluted cell suspensions were spotted on SD plates lacking uracil with or without 21 nM rapamycin and then grown at 30 °C for 3 (left) or 5 (right) days, respectively. **c** Pib2 active mutants enhance TORC1 activity in vitro. An in vitro TORC1 kinase assay was performed (as for Fig. 1d) using permeabilized yeast cells prepared from pib2Δ (MH1059) cells carrying a vector (p416ADH) or plasmids encoding wild-type *PIB2* (pMH330) or *PIB2* mutants (pMH334, pKY11-1, pKY14-1, pKY24-1, or pKY21). L.E long exposure, S.E short exposure. **d** The interaction between active Pib2 mutants and TORC1 is glutamine-responsive. Pull-down assays were performed using yeast cell lysates and bacterially-expressed GST-Pib2 tail motif mutants (as in Fig. 1b). The bar graph shows GST-Pib2-bound FLAG-Tor1 normalized to the wild-type Pib2 samples without glutamine. Error bars represent the standard deviation ($n = 3$ independent experiments).

which N-terminal Pib2 deletion confers cells with rapamycin resistance[19,20]. The N-terminal deletion Pib2 mutants (Δ1-115, Δ1-303) activated TORC1 in a glutamine-responsive manner, similar to wild-type Pib2 (Supplementary Fig. 1); therefore, we introduced random mutations starting from *PIB2(Δ1-115)* and screened for mutants that conferred higher rapamycin resistance than parental *PIB2(Δ1-115)* when transformed into pib2Δ cells. Interestingly, the majority of the isolated mutants had mutations

within the C-terminal tail motif (Fig. 3a, b). To examine whether these mutants enhanced TORC1 activity, we performed in vitro kinase assays using permeabilized cells expressing these *PIB2* mutants. We found that the cells expressing active mutants exhibited higher TORC1 activity than the wild-type cells in a glutamine-responsive manner (Fig. 3c), suggesting that the Pib2 tail motif is involved in TORC1 activation, but not primarily in glutamine-sensing. Next, the binding ability of the Pib2 tail motif

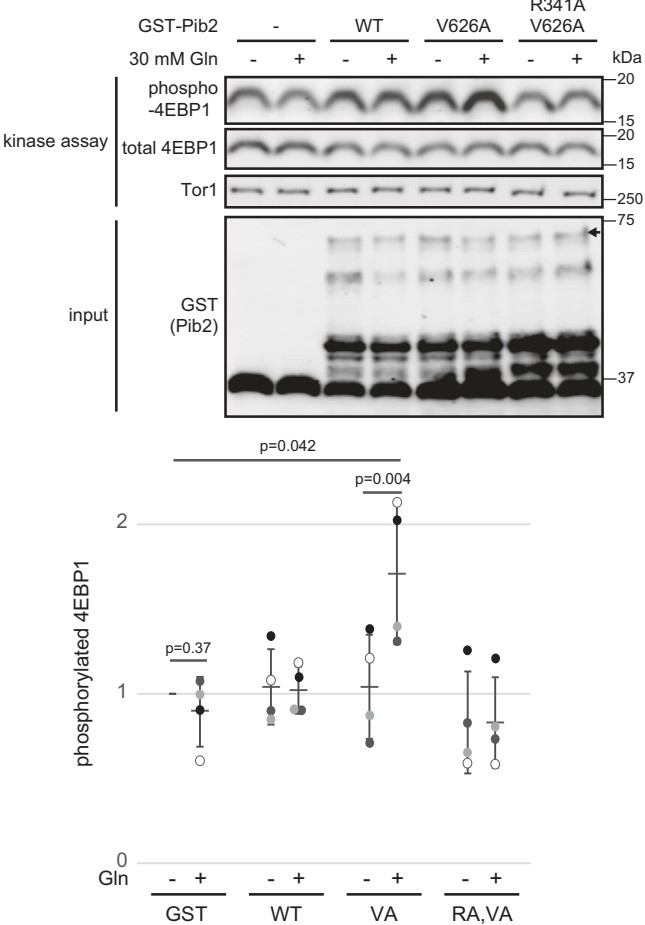

**Fig. 4 Pib2 directly activates TORC1 in vitro in response to glutamine.** In vitro kinase assays were performed using TORC1 purified from Kog1-Tap expressing cells and bacterially-expressed GST-Pib2(304-635) with or without 30 mM L-glutamine. Dot plots represent the ratio of phosphorylated/total 4EBP1 normalized to sample with GST but without glutamine. Error bars represent the standard deviation ($n = 4$ independent experiments), and significance was determined by a two-tailed Student's $t$-test.

mutants to TORC1 was tested by pull-down assay (Fig. 3d). The mutants also interacted with TORC1 in response to glutamine, although the interactions seemed slightly weakened compared to wild-type Pib2. This result suggests that the elevated TORC1 activation ability of the tail motif mutants is not due to increased binding to TORC1.

**Pib2 directly activates TORC1 in vitro in response to glutamine.** Next, we examined whether Pib2 directly activates TORC1 by performing in vitro kinase assays on purified TORC1 incubated with bacterially-expressed GST-Pib2(304-635) with or without glutamine. Although wild-type Pib2 barely enhanced TORC1 activity, the active Pib2(V626A) mutant obtained by screening (Fig. 3) only increased TORC1 activity in the presence of glutamine and this increase was reversed by the R341A mutation in the E motif (Fig. 4) which disrupted the glutamine-induced interaction between Pib2 and TORC1 (Fig. 2f). All other isolated tail motif mutants retained the ability to respond to glutamine when activating TORC1 (Supplementary Fig. 2). Together, these results indicate that Pib2 directly activates TORC1 in response to glutamine, but may require additional unidentified factor(s) for its full action since wild-type Pib2 did not significantly enhance TORC1 activity in vitro (Fig. 4).

**L-glutamine specifically changes Pib2 status.** Since our findings suggested that Pib2 has dual roles as a glutamine sensor and a direct TORC1 activator, we tried to detect the interaction between Pib2 and glutamine to test whether Pib2 acts as a glutamine sensor. The affinity between the assumed glutamine sensor and glutamine is predicted to be extremely low as the glutamine concentrations necessary for glutamine-dependent responses (Fig. 1c, d) were in the mM range, comparable to intracellular glutamine levels[3,22,23]. Accordingly, techniques commonly used to detect protein-ligand interactions, such as surface plasmon resonance or isothermal titration calorimetry, which require a higher affinity (Kd <1 mM and 100 μM, respectively), are not suitable for testing the interaction between Pib2 and glutamine. In addition, it was extremely difficult to purify Pib2 for structural analysis by methods such as X-ray crystallography and nuclear magnetic resonance because Pib2 has large intrinsically disordered regions (IDRs), observed as rope-like structures by high-speed atomic force microscopy (HS-AFM) (Supplementary Fig. 3), and was therefore highly susceptible to proteolytic degradation and aggregation. Consequently, we investigated the thermal stability of Pib2 in the presence or absence of glutamine using differential scanning calorimetry (DSC) to measure the enthalpic change (ΔH) of unfolding induced by heat. In this assay, we used bacterially-expressed thioredoxin-tagged Pib2(304-533, Δ356-384), which lacks the large IDRs and is therefore relatively resistant to degradation and aggregation but retains its glutamine-induced interaction with TORC1 (Fig. 5a, b). No significant shift in melting temperature was observed between the samples; however, the samples containing L-glutamine exhibited a higher ΔH than the samples without amino acids (Fig. 5c), whereas D-glutamine and L-asparagine decreased ΔH (Fig. 5c). We also confirmed that L-glutamine did not change the melting temperature or ΔH of thioredoxin (Fig. 5e). Together, these results indicate that L-glutamine specifically changes the folding state of Pib2. Next, the Pib2(R341A) mutant was subjected to DSC measurement (Fig. 5d). The result showed that a higher ΔH similar to wild-type Pib2 was specifically induced by L-glutamine. This result suggests that R341A is a mutant that retains glutamine-sensing but loses the TORC1-binding abilities.

## Discussion

In this study, we successfully reconstituted glutamine-responsive TORC1 activation via Pib2 in vitro and found that purified Pib2 and TORC1 were sufficient to induce the glutamine-responsive Pib2-TORC1 interaction and TORC1 activation. These observations suggest that Pib2 is the sole essential component of the Pib2 pathway and plays dual roles as a glutamine sensor and a direct TORC1 activator. In addition, our in vitro observations indicated that the vacuolar membranes on which both TORC1 and Pib2 reside are not required for the glutamine-induced interaction between Pib2 and TORC1 or for TORC1 activation, thus likely serves as a platform for the Pib2-TORC1 interaction in glutamine-responsive TORC1 activation rather than actively being involved in sensing or activation.

To appropriately respond to physiological changes in intracellular glutamine levels, the affinity between glutamine and the supposed sensor must be low since intracellular glutamine levels are high. Indeed, we found that mM glutamine concentrations comparable to intracellular levels are required to induce the Pib2-TORC1 interaction and in vitro Pib2-mediated TORC1 activation. Moreover, L-glutamine was found to induce marginal but significant changes in the folding state of Pib2, as indicated by DSC. L-glutamine, but not other amino acids, increased the Pib2 unfolding ΔH, likely by enhancing inter- or intra-molecular interactions.

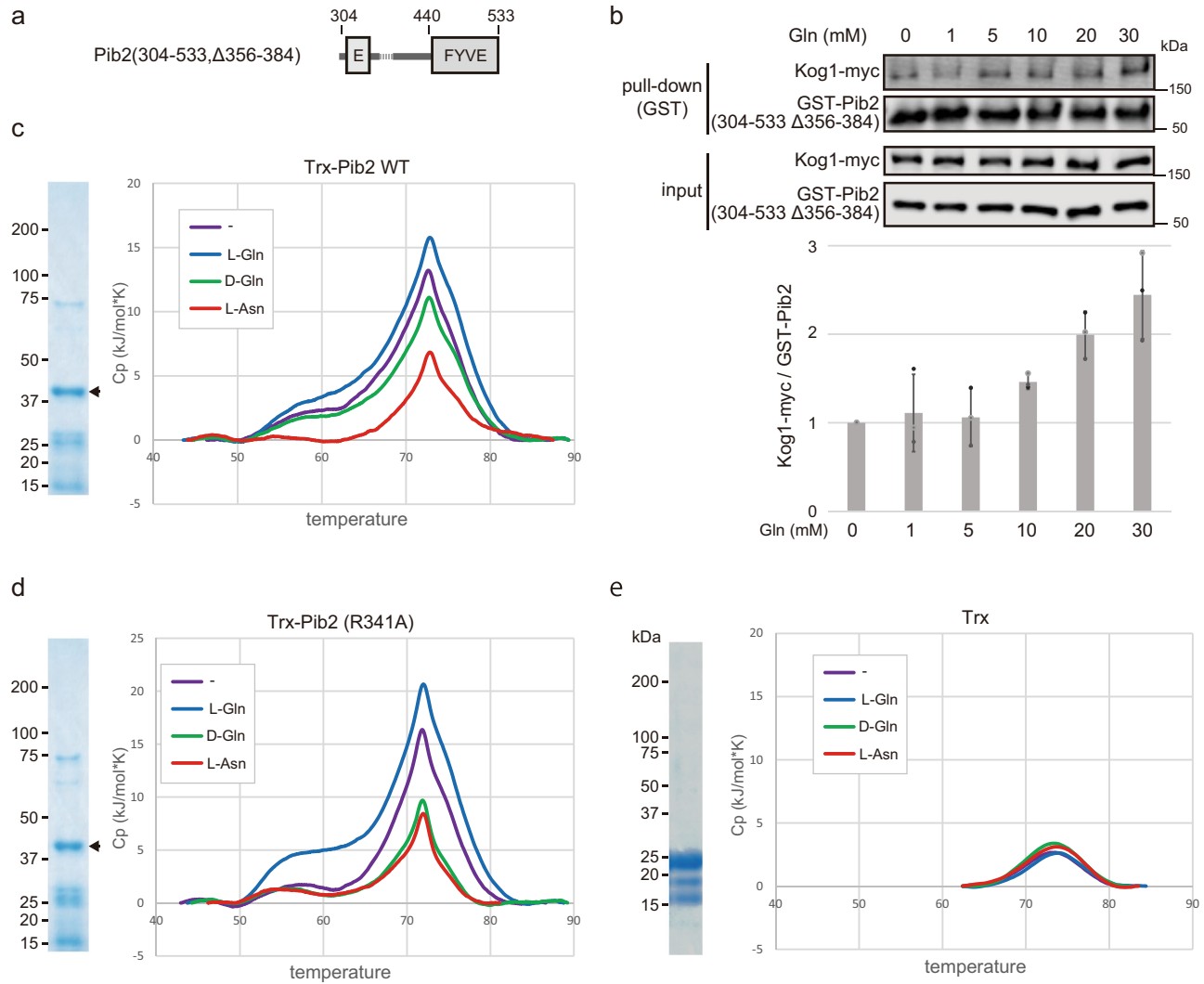

**Fig. 5 L-glutamine specifically changes Pib2 status. a** Schematic diagram of Pib2(304-533, Δ356-384) was used for DSC measurements. **b** Pib2(304-533, Δ356-384) interacts with TORC1 in response to glutamine. In vitro pull-down assays were performed (as in Fig. 1c) using GST-Pib2(304-533, Δ356-384) and yeast cell lysates prepared from Kog1-myc-expressing cells (TS270) with or without the indicated concentrations of L-glutamine. The line graph shows Kog1-myc bound to GST-Pib2(304-533, Δ356-384) normalized to sample without glutamine. The bar graph shows GST-Pib2-bound FLAG-Tor1 normalized to the samples without glutamine. Error bars represent the standard deviation ($n = 3$ independent experiments). **c, d** L-glutamine enhanced the ΔH of Pib2 heat-induced unfolding. **c** Thioredoxin (Trx)-tagged Pib2(304-533, Δ356-384) was subjected to DSC measurements with or without the indicated amino acids (20 mM). Purified samples were analysed by SDS-PAGE followed by CBB staining. **d** Trx-tagged Pib2(304-533, R341A, Δ356-384) was subjected to DSC measurements as in (**c**). **e** L-glutamine did not affect Trx, as determined by DSC measurements. Trx was subjected to DSC measurements as in (**c**).

Since the mutations in the independently isolated hyperactive *PIB2* mutants that still responded to glutamine converged in the C-terminal tail motif, this motif is likely to mechanistically induce TORC1 activation. On the basis of our data, we propose the following model of TORC1 activation (Fig. 6). First, glutamine binds to Pib2, which resides on the vacuolar membrane via its FYVE domain, and induces a conformational change in Pib2 which then interacts with TORC1 via its E motif. The association between Pib2 and TORC1 then makes it possible for the Pib2 tail motif to induce TORC1 activation. Our observations suggest that the E motif, FYVE domain, or both are the glutamine-sensing domain in Pib2 since Pib2(304-533, Δ356-384), which is composed mostly of the E and the FYVE domains, was found to exhibit a glutamine-responsive interaction with TORC1.

Although the Pib2 pathway is only activated by L-glutamine, an mM concentration is required, meaning that the Pib2-glutamine interaction is both specific and weak. This is an unusual feature of ligand-protein interactions, which generally display a positive correlation between specificity and affinity. Therefore, it remains unclear how Pib2 reconciles high specificity with low affinity. The IDRs of Pib2 may play an important role in this reconciliation by making Pib2 sway (Supplementary Fig. 3b), with this movement then promoting faster glutamine dissociation that results in low affinity. However, we do not propose that the IDRs directly bind to glutamine since their amino acid sequences are not conserved among Pib2 orthologues from other species.

Our findings also suggest that the Pib2 pathway is much simpler than the Gtr pathway, which involves numerous components. The simplicity and specificity of Pib2 for glutamine suggest that the Pib2 pathway is evolutionarily older than the Gtr pathway. Indeed, ancestral autotrophic organisms would not have had to monitor amino acid levels as they could coordinately synthesize all proteinogenic amino acids; therefore, it is reasonable to assume that they monitored glutamine, one of the most

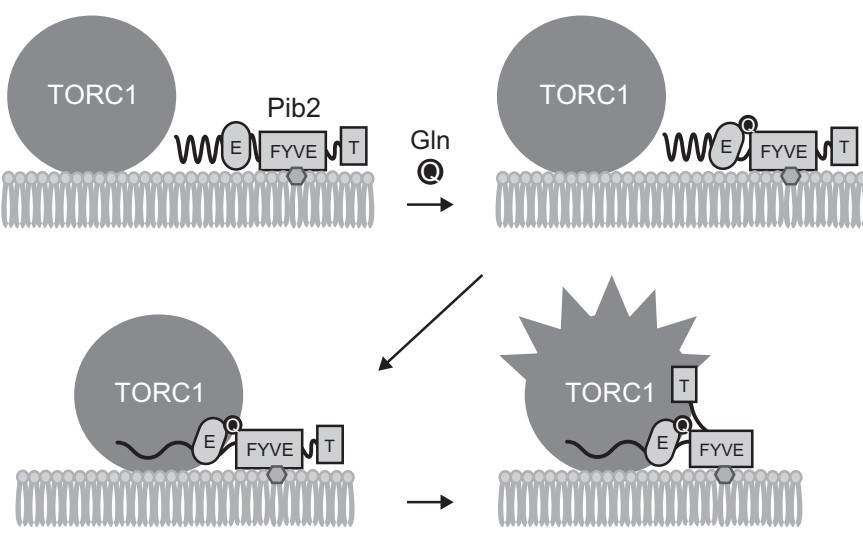

**Fig. 6 A model of Pib2-dependent TORC1 activation in response to glutamine.** See the discussion section for details.

| Table 1 Yeast strains used in this study. | | |
|---|---|---|
| **Strain** | **Genotype** | **Reference or source** |
| TM101 | *MATα leu2-Δ1 his3-Δ200 ura3-52* | 29 |
| TM141 | *MATa leu2-Δ1 his3-Δ200 trp1-Δ63 ura3-52* | 29 |
| RL171-2a | *MATα leu2-3,112 trp1 ura3-52 his3 rme1 KOG1-TAP::His3MX6* | 27 |
| MH1059 | Same as TM141 except *pib2::kanMX4* | 18 |
| MH1030 | Same as TM101 except *gtr1::hphNT1* | This study |
| TS269 | *MATα leu2-Δ1 his3-Δ200 ura3-52 3FLAG-TOR1::His3MX6* | Lab stock |
| TS270 | *MATa leu2-Δ1 his3-Δ200 ura3-52 3FLAG-TOR1::His3MX6 KOG1-13myc::kanMX4* | Lab stock |
| KY3 | Same as TM141 except *rtg2::kanMX4* | This study |
| KY13 | *MATα leu2-Δ1 his3-Δ200 ura3-52 rtg2::kanMX4 gtr1::hphNT1* | This study |

abundant amino acids, to indicate cellular nitrogen status. Once heterotrophic organisms evolved, some lost certain amino acid synthesis pathways and essential amino acids emerged, necessitating a system to monitor their availability. For instance, leucine and arginine are essential or semi-essential amino acids in mammals, respectively, that have specific sensor molecules to monitor their levels. The Pib2 pathway may also be conserved in other species, since mammals and plants have Pib2 homologs known as PLEKHF1/LAPF/phafin-1/ZFYVE15 and FYVE1/FREE1, respectively[17,24]. Interestingly, the Rag/Gtr complex is not conserved in plants; therefore, the Pib2 pathway may play a predominant role in TORC1 activation in plants.

In conclusion, this study demonstrated that Pib2 is a glutamine sensor that directly activates TORC1. To further elucidate the detailed mechanism underlying sensing and activation during glutamine-responsive TORC1 activation, structural analyses are required.

## Methods

**Yeast strains and plasmids.** The yeast strains and plasmids used in this study are listed in Tables 1 and 2, respectively.

Strain RL171-2a was a kind gift from Dr. R. Loewith, while strains MH1030 and KY3 were constructed by replacing *GTR1* in TM101 and *RTG2* in TM141 with the *hphNT1* and *kanMX4* cassettes, respectively, and strain KY13 was obtained by

crossing strains MH1030 and KY3. To create pMH330, the *PIB2* ORF was cloned into p416ADH[25], while pMH342 and pMH433 were created by cloning a genomic *PIB2* fragment containing promoter and terminator regions into pRS416 and pRS415[26], respectively. The *pib2(P337A)*, *pib2(L340A)*, and *pib2(R341A)* mutants were constructed by site-directed mutagenesis. Fragments encoding *PIB2* deletion mutants were cloned into p416ADH, pET48b(+), pET49b(+), and pET50b(+). To create pMH649, a thioredoxin-hexahistidine-*PIB2 (304-635)* fragment was cloned into pTBK1. pKY series plasmids were obtained by hyperactive mutant screening.

**Reagents.** Anti-Flag M2 was purchased from Sigma-Aldrich (St. Louis, MO, USA). Anti-Tor1 (yN-15) was purchased from Santa Cruz Biotechnology (Dallas, TX, USA). Anti-myc (9E10) was a kind gift from Dr. Hoshikawa. Anti-phospho-4EBP1 (T37/46) (#9459) and anti-4EBP1 (#9452) were purchased from Cell Signaling Technology (Danvers, MA, USA). Anti-GST (4C10) was purchased from Abcam (Cambridge, UK). Anti-Penta His was purchased from QIAGEN (Hilden, Germany). IRdye800-conjugated anti-rabbit antibody was purchased from LI-COR (Lincoln, NE, USA). Alexa 680-conjugated anti-mouse and anti-goat antibodies were purchased from Thermo Fisher Scientific (Waltham, MA, USA).

**GST-Pib2 purification from *E. coli*.** *E. coli* BL21 CodonPlus RIL carrying pET49(+)-derived plasmids were cultivated in LB containing 30 μg mL$^{-1}$ kanamycin and 10 μg mL$^{-1}$ chloramphenicol at 37 °C overnight. The overnight culture was then diluted in LB containing 30 μg mL$^{-1}$ kanamycin to an OD$_{600}$ of 0.35 and incubated for 30 min at 37 °C before 0.5 mM of isopropyl-β-D-thiogalactoside (IPTG) was added. The culture was then incubated for a further 3.5–4 h at 24 °C and the cells were collected, suspended in PBSN buffer (PBS + 0.1% NP40), and lysed by sonication. After centrifugation at 15,000× *g* for 10 min, the supernatant was mixed with glutathione Sepharose 4B beads (GE Healthcare Life Sciences, Amersham, UK) and the mixture was rotated for 1 h at 4 °C. The beads were washed three times with PBSN and precipitated proteins were eluted in elution buffer (50 mM Tris-HCl pH 8.0, 10 mM reduced glutathione).

**Western blotting.** To quantify protein levels, fluorescent dye-conjugated secondary antibodies were used and the fluorescence was measured using an Odyssey scanner (LI-COR). Band intensity was quantified using Image Studio ver.4.0 (LI-COR Biosciences). Standard curves of fluorescence intensity versus sample quantity were produced for each blotted membrane for quantification.

**In vitro pull-down assay using cell lysates.** TS269 (*3FLAG-TOR1*) or TS270 (*3FLAG-TOR1 KOG1-13myc*) cells were cultured in YPD medium to an OD$_{600}$ of 2.0–3.0 before being collected by centrifugation, suspended in lysis buffer A (40 mM HEPES-KOH [pH 7.5], 120 mM NaCl, 1 mM EDTA, 50 mM NaF, 0.3% CHAPS, 1 mM phenylmethylsulfonyl fluoride, 40 μg mL$^{-1}$ of aprotinin, 10 μg mL$^{-1}$ of pepstatin A, 20 μg mL$^{-1}$ of leupeptin), and disrupted using a Multi-Beads Shocker (Yasui Kikai, Osaka, Japan) with 0.5 mm zirconia beads (Nikkato, Osaka, Japan). After the samples had been centrifuged at 12,000 × *g* for 15 min at 4 °C, the supernatants were collected, the indicated amount of each amino acid was added alongside purified GST-Pib2, and the mixture was incubated for 1 h on ice. Glutathione Sepharose beads were then added and the mixture was rotated for 1 h at 4 °C. After the beads had been washed three times with lysis buffer A containing each amino acid at the indicated amount, the beads were suspended in Laemmli

**Table 2 Plasmids used in this study.**

| Plasmids | Description | Reference or source |
|---|---|---|
| pRS416 | Yeast shuttle vector | 26 |
| p416ADH | Yeast shuttle vector | 25 |
| p425ADH | Yeast shuttle vector | 25 |
| pET48b(+) | Trx-His tagged protein expression vector | Merck Millipore |
| pET49b(+) | GST-tagged protein expression vector | Merck Millipore |
| pET50b(+) | His-Nus-His tagged protein expression vector | Merck Millipore |
| pTBK1 | Intein-CBD tagged protein expression vector | New England Biolabs |
| pMH330 | p416ADH-*PIB2* | This study |
| pMH334 | p416ADH-*pib2 (Δ1-115)* | This study |
| pMH342 | pRS416-*PIB2* | This study |
| pMH410 | p416ADH-*pib2 (Δ1-303)* | This study |
| pMH417 | pET49b(+)-*pib2 (304-635)* | This study |
| pMH422 | pET49b(+)-*pib2 (304-620)* | This study |
| pMH426 | pET49b(+)-*pib2 (440-635)* | This study |
| pMH433 | pRS415-*PIB2* | This study |
| pMH497 | pRS416-*pib2(P337A)* | This study |
| pMH498 | pRS416-*pib2(R341A)* | This study |
| pMH499 | pRS416-*pib2(L340A)* | This study |
| pMH508 | Same as pMH417 except *PIB2(V626A)* | This study |
| pMH510 | p416ADH-*pib2(P337A)* | This study |
| pMH512 | p416ADH-*pib2(L340A)* | This study |
| pMH514 | p416ADH-*pib2(R341A)* | This study |
| pMH553 | Same as pMH417 except *PIB2(P627S)* | This study |
| pMH554 | Same as pMH417 except *PIB2(A628T)* | This study |
| pMH555 | Same as pMH417 except *PIB2(N629S)* | This study |
| pMH568 | Same as pMH417 except *(R341A, V626A)* | This study |
| pMH606 | pET48b(+)-*pib2(304-533, Δ356-384)* | This study |
| pMH610 | pET50b(+)-*pib2(304-533, Δ356-384)* | This study |
| pMH649 | pTBK1-Trx-His-pib2(304-635) | This study |
| pKY11-1 | Same as pMH334 except *PIB2(V626A)* | This study |
| pKY14-1 | Same as pMH334 except *PIB2(P627S)* | This study |
| pKY24-1 | Same as pMH417 except *PIB2(A628T)* | This study |
| pKY21 | Same as pMH334 except *PIB2(N629S)* | This study |
| pKY95 | Same as pMH342 except *pib2(Δ356-384)* | This study |

sample dye and boiled for 4 min. The supernatants were subjected to western blotting and the blotted bands were quantified as described above.

**In vitro kinase assay using permeabilized cells.** Logarithmically growing cells in YPD medium were harvested by centrifugation and washed with MilliQ water containing 2 mM phenylmethylsulfonyl fluoride (PMSF). The cells were suspended in DTT solution (0.1 M Tris-HCl (pH 9.4), 10 mM DTT) and incubated for 10 min at room temperature. Then, the cells were collected, suspended in spheroplasting buffer (0.7 M sorbitol, 10 mM Tris-HCl [pH 7.5], 1 mM DTT, 20 mM NaN$_3$, and 0.15 mg ml$^{-1}$ Zymolyase-100T), and incubated for 15–20 min at room temperature. The spheroplasted cells were washed once with 1 ml of cold sorbitol buffer (1 M sorbitol, 150 mM K-acetate, 5 mM Mg-acetate, 20 mM HEPES-KOH [pH 6.6]), resuspended in the same buffer to an optical density of 100 at 600 nm (OD600) mL$^{-1}$, and stored at −80 °C in small aliquots. The cells were thawed at 30 °C for 1 min, diluted with 5 times the volume of buffer C (0.25 M sorbitol, 150 mM potassium acetate, 5 mM acetate, 20 mM HEPES-KOH [pH 6.6], 1 mM phenylmethylsulfonyl fluoride, 40 µg mL$^{-1}$ aprotinin, 10 µg mL$^{-1}$ pepstatin A, 20 µg mL$^{-1}$ leupeptin fluoride), and incubated on ice for 5 min. The cell mixture was centrifuged at 1000× g for 2 min at 4 °C, and the cell pellet was resuspended in buffer C. The permeabilized cells were obtained by repeating the washing treatment with buffer C, with buffer D (same as buffer C except for 0.05 M sorbitol), and then with buffer C, in turn. Then the permeabilized cells were suspended in buffer E (0.4 M sorbitol, 150 mM K-acetate, 5 mM Mg-acetate, and 20 mM HEPES-KOH [pH 6.6]). Cells (0.7 OD600) were then subjected to the kinase reaction in reaction buffer (0.4 M sorbitol, 150 mM K-acetate, 5 mM Mg-acetate, 20 mM HEPES-KOH [pH 6.6], 40 mM creatine phosphate, 200 ng µL$^{-1}$ creatine kinase, 1 mM Pefabloc SC, 4 µg mL$^{-1}$ aprotinin, 1 µg mL$^{-1}$ pepstatin A, 2 µg mL$^{-1}$ leupeptin, 10 µg mL$^{-1}$ GST-4EBP1, 0.5 mM ATP, indicated amount of glutamine). After incubation for 10 min at 30 °C, the kinase reaction was stopped by adding Laemmli sample buffer and the sample was boiled for 4 min before being centrifuged at 9000× g for 2 min. The supernatant was subjected to western blotting with anti-phospho-4EBP1 or anti-GST antibodies and the blotted bands were quantified as described above.

**Trx-His-Pib2(304-635) purification using an intein-mediated purification system.** Trx-His-Pib2(304-635) was purified using an IMPACT™ system (New England Biolab, Ipswich, MA, USA) according to the manufacturer's instruction. Briefly, *E. coli* BL21 CodonPlus RIL carrying pMH649 was cultivated in LB containing 100 µg mL$^{-1}$ ampicillin and 10 µg mL$^{-1}$ chloramphenicol at 37 °C overnight. The overnight culture was diluted in LB containing 100 µg mL$^{-1}$ ampicillin to an OD$_{600}$ of 0.35, incubated for 30 min at 37 °C, and 0.5 mM of IPTG was added before further incubation for 3.5 h at 24 °C. The cells were then collected and suspended in chitin-binding buffer (20 mM HEPES-KOH [pH 7.5], 1 M NaCl, 0.2% Triton X-100, 5% glycerol, 1 mM phenylmethylsulfonyl fluoride, 40 µg mL$^{-1}$ aprotinin, 10 µg mL$^{-1}$ pepstatin A, 20 µg mL$^{-1}$ leupeptin), and lysed by sonication. After centrifugation at 15,000× g for 15 min, the supernatant was mixed with chitin beads (New England Biolabs), and the mixture was rotated for 1 h at 4 °C. The beads were washed three times with chitin-binding buffer and on-bead self-cleavage was induced in cleavage buffer (20 mM HEPES-KOH [pH 7.5], 1 M NaCl, 0.2% Triton X-100, 5% glycerol, 50 mM DTT) at 4 °C overnight. After being centrifuged at 2500× g for 30 sec, supernatants containing Trx-His-Pib2(304-635) were collected and dialyzed in dialysis buffer 1 (50 mM Tris-HCl [pH 7.5], 1 M NaCl, 0.2% NP-40, 10% glycerol, 50 µM ZnCl$_2$, 1 mM DTT).

**In vitro pull-down assay using purified TORC1.** TORC1 was purified from Kog1-TAP-expressing yeast (RL171-2a) as described previously[27]. Briefly, purified Kog1-TAP on magnetic beads was suspended in pull-down buffer (50 mM Tris-HCL[pH 7.5], 200 mM NaCl, 5% glycerol, 0.1% NP-40, 0.5 mM DTT, 1 mM phenylmethylsulfonyl fluoride, 40 µg mL$^{-1}$ aprotinin, 10 µg mL$^{-1}$ pepstatin A, 20 µg mL$^{-1}$ leupeptin), mixed with Trx-His-Pib2(304-635), and purified using an IMPACT™ system. After the mixture had been rotated for 2 h at 4 °C, the beads were washed three times with pull-down buffer, suspended in Laemmli sample dye, and boiled for 4 min. Pull-down assays were also performed in the same manner, except the pull-down buffer was replaced with lysis buffer A and Trx-His-Pib2 was replaced by GST-Pib2(304-635). In another variation, Kog1-TAP was released from magnetic beads by tobacco etch virus protease cleavage and the cleaved fraction was subjected to pull-down with GST-Pib2 in lysis buffer A.

**Active *PIB2* mutant screening**. *pib2(Δ1-115)* was randomly mutagenized by error-prone PCR using ExTaq (Takara Bio, Shiga, Japan) according to the manufacturer's instructions but using twice the amount of dNTP (5 mM each). PCR products were introduced into *gtr1Δ rtg2Δ* (KY13) together with linearized pMH334 and the transformants were plated on SD medium supplemented with leucine and histidine, on which the parental strain (KY13) cannot grow. Candidate plasmids were isolated from the colonies that grew on the plates and introduced into the *gtr1Δ* strain (MH1030). Rapamycin resistance in the transformants was examined.

**In vitro kinase assay using purified TORC1**. Purified TORC1 on magnetic beads was prepared as described above and suspended in lysis buffer A ± 30 mM L-glutamine. Wild-type GST-Pib2(304-635) or the indicated mutants purified from *E. coli* were added to the samples and incubated for 2 h at 4 °C. The beads were briefly washed with the same buffer and then suspended in kinase buffer (0.4 M sorbitol, 150 mM K-acetate, 5 mM Mg-acetate, 20 mM HEPES-KOH [pH 6.6], 40 mM creatine phosphate, 200 ng μL$^{-1}$ creatine kinase, 1 mM Pefabloc SC, 4 μg mL$^{-1}$ aprotinin, 1 μg mL$^{-1}$ pepstatin A, 2 μg mL$^{-1}$ leupeptin, and 13 μg mL$^{-1}$ 4EBP1) ± 30 mM L-glutamine. Kinase reactions were started by adding 0.5 mM of ATP, followed by incubation for 8 min at 30 °C, and then stopped by adding Laemmli sample buffer boiled for 4 min, and subjected to western blotting. Data represent four independent experiments and were quantified as described above. Statistical analyses were performed using Student's *t*-tests.

**Purification of Trx-His- and His-Nus-His-tagged proteins from *E. coli* by nickel-affinity chromatography**. *E. coli* BL21 CodonPlus RIL cells carrying pET48b(+), pMH606, pET50b(+), or pMH610 were cultivated in LB containing 30 μg mL$^{-1}$ kanamycin and 10 μg mL$^{-1}$ chloramphenicol at 37 °C overnight. The overnight culture was diluted to an OD$_{600}$ of 0.35 in LB containing kanamycin and incubated for 20 min at 37 °C and a further 10 min at 24 °C. Next, 0.5 mM IPTG was added, and the culture was incubated for 3.5 h at 24 °C before the cells were collected and suspended in His-purification buffer (50 mM Tris-HCl (pH7.5), 1 M NaCl, 0.1% NP-40, 10% glycerol, 20 mM imidazole). Lysozyme (1 μg mL$^{-1}$) was then added and the mixture was incubated for 30 min on ice before the samples were lysed by sonication. After centrifugation at 15,000 × *g* for 20 min, the supernatant was mixed with Ni-NTA agarose beads (QIAGEN), rotated for 0.7 h at 4 °C, and the beads were washed with His-purification buffer three times. Precipitated proteins were eluted using elution buffer (same as His-purification buffer except for 200 mM imidazole) and dialyzed in dialysis buffer 2 (50 mM Tris-HCl (pH6.8), 1 M NaCl, 10% glycerol, 0.2% NP-40, 0.5 mM DTT).

**DSC measurements**. Bacterially-expressed Trx-His-tagged samples purified by nickel-affinity chromatography were dialyzed in dialysis buffer 3 (25 mM HEPES-KOH (pH7.5), 500 mM NaCl, 10% glycerol, 0.1% NP-40). Protein samples (250–350 ng μL$^{-1}$) with or without the indicated amino acids (20 mM) were subjected to MicroCal PEAQ-DSC automated analysis (Malvern Panalytical, Malvern, United Kingdom).

**HS-AFM imaging of Pib2**. HS-AFM images were acquired in tapping mode using a tip-scan type HS-AFM instrument (Nano Explorer PS-NEX, Research Institute of Biomolecule Metrology, Ibaraki, Japan). His-Nus-His-tagged Pib2(304-533, Δ356-384) (~1 nM) in 20 μL of 20 mM HEPES-NaOH pH 8.0 containing 300 mM NaCl was deposited onto freshly cleaved mica (~3 × ~3 mm$^2$) attached to a coverslip (24 × 32 mm$^2$, 0.13–0.17 mm-thick; Matsunami Glass, Osaka, Japan). After incubation at RT for 5 min, the excess protein was washed away using observation buffer (20 mM HEPES-NaOH pH 8.0 containing 20 mM NaCl). We used cantilevers (~9 μm long, ~2 μm wide, ~0.13 μm thick) with a resonant frequency of ~1.5 MHz and a spring constant of 0.1–0.2 N m$^{-1}$ (BL-AC10DS, Olympus). Imaging conditions were as follows: scan size, 150 × 60 nm$^2$; pixel size, 120 × 48 pixels; imaging rate, 5 frames sec$^{-1}$. Imaging was performed at 23 °C.

**Statistics and reproducibility**. Statistical analyses were performed from at least three independent experiments. Specific information on the number of replicates and statistical analysis is included in each figure legend. The *P* values <0.05 were considered significant and shown in figures.

**Reporting summary**. Further information on research design is available in the Nature Research Reporting Summary linked to this article.

## Data availability

The datasets generated during and/or analysed during the current study are available in the figshare repository, https://doi.org/10.6084/m9.figshare.14891628[28]. Uncropped and unedited blot/gel images are also included at the end of the Supplementary Figures file. Any remaining information can be obtained from the corresponding author upon reasonable request.

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

## Acknowledgements

We thank Robbie Loewith and Yutaka Hoshikawa for reagents used in this study, Terunao Takahara for his constructive comments on the manuscript, Advanced Research

Facilities and Services, HUSM for technical support and allowing us to use the facility, and all members of Maeda lab and Hiroyuki Tachikawa lab for help and discussion. This work was supported in part by JSPS KAKENHI Grant Numbers 20K06555 (to M.T.), 18H02151 (to K.N.), 19K16344 (to D.N.), 19H05707, 18H03989 (both to N.N.N), 17H03802, and 20H03251 (both to T.M.), Platform Project for Supporting Drug Discovery and Life Science Research (Basis for Supporting Innovative Drug Discovery and Life Science Research (BINDS)) from AMED under Grant Number JP19am0101094 (support number 1995) (to K.T.), grants from Ohsumi Frontier Science Foundation (to T.M.) and from Hamamatsu Foundation for Science and Technology Promotion (to M.T.), and HUSM Grant-in-Aid (to T.M. and M.T.).

## Author contributions

Conceptualization, M.T., and T.M.; Methodology, M.T., S.N., K.N., N.N.N., K.T., and T.M.; Investigation, M.T., K.Y., S.N., K.N., and D.N.; Writing – Original Draft, M.T. and T.M.; Writing – Review & Editing, M.T., and T.M.; Funding Acquisition, M.T., K.N., D.N., N.N.N., K.T., and T.M.; Supervision, S.N., K.N., N.N.N., K.T., and T.M.

## Competing interests

The authors declare no competing interests.
