## [Peer Review File · Communications Biology]

Reviewers' comments:

Reviewer #1 (Remarks to the Author):

The manuscript by Tanigawa et al. focuses on Pib2, a vacuolar protein previously proposed as glutamine sensor for TORC1 signaling. In this small study, the authors confirmed some previously published data and investigated more the mechanism by which Pib2 interacts and regulates TORC1 in response to glutamine. While being short, this study is interesting and relevant for the TORC1 field. However, I have few questions and remarks about some of the claims and experiments.

Figure 1D: It is unclear how this assay is specifically monitoring Pib2-dependent TORC1 activation since permeabilized yeast cells are used. Other vacuolar proteins may remain in the permeabilized yeast cells and therefore influence TORC1 activation in response to glutamine. Can the authors clarify this point?

Figure 2B: The authors claimed that Pib2(304-620) retained the ability to interact with TORC1 in response to glutamine suggesting that the tail motif is dispensable for the glutamine-induced Pib2-TORC1 interaction. It is true that the Pib2(304-620)-TORC1 interaction seems slightly induced by glutamine. However, both basal (-glutamine) and glutamine stimulated interactions are strongly reduced compared to the Pib2(304-635). To further support their claim, the authors should quantify the levels of FLAG-Tor1 after pull down and see if the fold induction mediated by glutamine is the same between the Pib2(304-635) and (304-620)

Figure 2DEF: While partially rescuing rapamycin sensitivity and TORC1 signaling (Fig 2DE), the L340A mutant does not interact with TORC1 (Fig 2F) suggesting that the L340A mutant regulates TORC1 independently of the Pib2-TORC1 interaction. Can the authors explain and discuss this result?

Figure 2E: The quantification displayed here represents only one experiment. Quantifying 3 independent experiments would be more relevant and give more strength to the figure.

Figure 3 and 4: Based on Fig 2B, the Pib2 tail motif is required for at least the basal interaction between Pib2 and TORC1. Therefore, the author should measure the effect of these mutation on the basal and glutamine induced Pib2-TORC1 interaction.

Reviewer #2 (Remarks to the Author):

This manuscript extends previous studies by showing that yeast Pib2 proteins binds L-glutamine and then binds and activates TORC1 in response to L-glutamine in vitro. Two conserved domains were found to be required for these interactions. First, the E-domain was critical for glutamine-dependent association with, and activation of, purified TORC1. Second, in an unbiased mutagenesis approach, several amino acids within the tail-domain of Pib2 seemed to hyperactivate its effects on TORC1. A model consistent with the findings is also presented. These findings are well-supported with careful experiments and represent significant advances in the field. There are a few minor issues that should be addressed, however, with at least one major concern as follows.

Major concerns

1. Interesting data were provided using differential scanning calorimetry that Pib2 undergoes folding (increased heat capacity) upon binding L-glutamine. However, L-asparagine appears to produce a very strong effect (Fig 5C) that is even stronger when normalized by peak Cp's in the Trx domain. These data were described inaccurately in the Results as decreasing ΔH when it actually increased (Line 182). These findings raise the question of whether L-asparagine could bind Pib2 and interfere with the positive effects of L-glutamine in the binding and activation assays of TORC1, which would suggest that Pib2 senses glutamine/asparagine ratios rather than glutamine itself. More experiments may be necessary to resolve this question.

2. Additionally, can the DSC experiment be replicated using instead the R341A variant of Pib2,

which is reported here as critical for L-glutamine responsiveness? This simple experiment has the potential of resolving the issue above regarding L-asparagine as well as significantly increasing the resolution of the model.

Minor concerns

1. No clear orthologs of Pib2 have been reported in plants, and there is no data to suggest that the *Arabidopsis thaliana* protein included in the multiple sequence alignments (Fig 2C, 3A) interacts in any way with TORC1. Unless there is more data to support the inclusion of the plant sequences, they can be removed from the figures and the Discussion can be corrected without any loss of clarity.

2. Similarly, the E-domain of Pib2 is not conserved in *S. pombe* and by including this sequence (along with that of *A. thaliana*) in Fig 2C the color scheme used to represent conservation has been substantially altered. Both non-homologous sequences should be deleted from Fig 2C unless more evidence can be supplied regarding conservation.

Reviewer #3 (Remarks to the Author):

The manuscript by Tanigawa et al. investigates the role of Pib2 in the glutamine-dependent upstream regulation of TORC1 in budding yeast. The authors' aim is to gain more insights into the signal transduction of glutamine to TORC1 via direct binding to Pib2, which was previously shown in papers from their own group and others (Tanigawa and Maeda, 2017; Ukai et al., 2018). The characterization of the Pib2-dependent activation of TORC1 by L-glutamine is indeed of great interest for the community working on TORC1 upstream regulation. The manuscript is well written and the conclusions seem overall sound. However, some general technical aspects and specific issues would need to be addressed to properly back the claims of the paper.

A general remark: standard deviation bars (and the relative statistics) are represented only in Fig. 4, while I feel that this would be needed elsewhere in the paper, specifically where some graphs represent quantifications of blots where differences are hard to see by eye (see the specific comments below, this is especially needed in Fig. 5B).

Specific comments:

- Lines 102-105. The authors show side-by-side pull-downs and in vitro kinase assays with the same glutamine concentrations (Fig. 1C-D), which make sense. However, Fig. 1C is basically the same as in a previous paper (Ukai et al., 2018, Fig. 5B). Therefore, I propose to mention it in the text ("In agreement with...").
- Lines 119-120. "Pib2(304-620), which lacks the C-terminal tail motif essential for TORC1 activation ^{17,19,20}, still interacted with TORC1 in response to glutamine (Fig. 2B),...". This claim seems not to be supported by the blot, since in the pull-down of this Pib2 fragment both FLAG-Tor1 bands (with and without glutamine) are less intense than the respective controls (GST mock pull-downs).
- Lines 176-179. "In this assay, we used bacterially-expressed thioredoxin-tagged Pib2(304-533, Δ356-384), which lacks the large IDRs and is therefore relatively resistant to degradation and aggregation but retains its glutamine-induced interaction with TORC1 (Fig. 5A,B)". The claimed 6-fold induced interaction in the presence of 30 mM glutamine is absolutely not observable from the blot. As abovementioned, this is the most relevant case where SDs (and maybe another blot) are needed in order to back the claim.

Reviewer #1 (Remarks to the Author):

The manuscript by Tanigawa et al. focuses on Pib2, a vacuolar protein previously proposed as glutamine sensor for TORC1 signaling. In this small study, the authors confirmed some previously published data and investigated more the mechanism by which Pib2 interacts and regulates TORC1 in response to glutamine. While being short, this study is interesting and relevant for the TORC1 field. However, I have few questions and remarks about some of the claims and experiments.

We are encouraged to know that you deemed the observation dealt with in our manuscript interesting. We also thank you for the insightful comments, which are helpful to improve our original manuscript. We have sincerely revised the manuscript in line with the helpful suggestions, hoping that you find the revised version adequately improved.

Figure 1D: It is unclear how this assay is specifically monitoring Pib2-dependent TORC1 activation since permeabilized yeast cells are used. Other vacuolar proteins may remain in the permeabilized yeast cells and therefore influence TORC1 activation in response to glutamine. Can the authors clarify this point?

Upon this comment, we have realized that our original description was not clear enough regarding the specificity of the assay used in Fig.1D. In our previous report (doi: 10.1128/MCB.00075-17), we have shown that the assay using permeabilized cells monitors only the Pib2-dependent fraction of the TORC1 activation since the samples prepared from *pib2Δ* cells do not respond to glutamine. Similar results are also shown in Fig.2e of the current manuscript. To clarify this point, we have added the following explanation (lines 100-103).

“To measure TORC1 activity, we used an in vitro kinase assay using permeabilized yeast cells as the TORC1 kinase source¹⁸. The assay monitors only Pib2-dependent TORC1 activation since permeabilized cells prepared from pib2Δ cells do not respond to glutamine¹⁸.”

Figure 2B: The authors claimed that Pib2(304-620) retained the ability to interact with TORC1 in response to glutamine suggesting that the tail motif is dispensable for the glutamine-induced Pib2-TORC1 interaction. It is true that the Pib2(304-620)-TORC1 interaction seems slightly induced by glutamine. However, both basal (-glutamine) and

glutamine stimulated interactions are strongly reduced compared to the Pib2(304-635). To further support their claim, the authors should quantify the levels of FLAG-Tor1 after pull down and see if the fold induction mediated by glutamine is the same between the Pib2(304-635) and (304-620).

We agree that the glutamine-responsive TORC1-Pib2(304-620) interaction is not dramatic. Thus, as you suggested, we added quantified results in Fig.2b. As you pointed out, both the basal and the glutamine-stimulated TORC1-Pib2(304-620) interactions were weaker than those of TORC1-Pib2(304-635). Nonetheless, the fold-induction mediated by glutamine was comparable between Pib2(304-635) and Pib2(304-620), supporting our original claim that the tail motif is dispensable for the glutamine-responsiveness of the Pib2-TORC1 interaction. For clarity, we also revised the corresponding clause as follows: “, *indicating that the tail motif is dispensable for the glutamine- responsiveness of the Pib2-TORC1 interaction*” (line 122-124).

Figure 2DEF: While partially rescuing rapamycin sensitivity and TORC1 signaling (Fig 2DE), the L340A mutant does not interact with TORC1 (Fig 2F) suggesting that the L340A mutant regulates TORC1 independently of the Pib2-TORC1 interaction. Can the authors explain and discuss this result?

Thank you for providing the insight. However, considering the rather weak signal of Pib2(P337) in Fig.2f, and more importantly, considering the consistent ordering among the mutants (i.e. WT>P337A>L340>R341A) in the three different assays shown in Fig.2d, e, f, it seems more likely that the signal for Pib2(L340) is undetectably marginal although the Pib2 mutants' potential to activate TORC1 is dictated by the strength of their interaction with TORC1.

Figure 2E: The quantification displayed here represents only one experiment. Quantifying 3 independent experiments would be more relevant and give more strength to the figure.

We added the quantified results from three independent experiments in Fig2e.

Figure 3 and 4: Based on Fig 2B, the Pib2 tail motif is required for at least the basal interaction between Pib2 and TORC1. Therefore, the author should measure the effect of these mutation on the basal and glutamine induced Pib2-TORC1 interaction.

Thank you for the thoughtful suggestion. We performed the pull-down assay using the Pib2 hyperactive mutants and added the results to our current manuscript as Fig.3d (lines 150-154). The mutants also interacted with TORC1 in response to glutamine, although the interaction seemed slightly weakened compared to wild-type Pib2. At present, we cannot clearly explain why the tail mutants weakly interact but strongly activate TORC1, but speculate that the TORC1 activation by the tail motif requires not only binding but other regulatory steps, which are enhanced in the mutants. We are currently pursuing the underlying mechanism.

Reviewer #2 (Remarks to the Author):

This manuscript extends previous studies by showing that yeast Pib2 proteins binds L-glutamine and then binds and activates TORC1 in response to L-glutamine in vitro. Two conserved domains were found to be required for these interactions. First, the E-domain was critical for glutamine-dependent association with, and activation of, purified TORC1. Second, in an unbiased mutagenesis approach, several amino acids within the tail-domain of Pib2 seemed to hyperactivate its effects on TORC1. A model consistent with the findings is also presented. These findings are well-supported with careful experiments and represent significant advances in the field. There are a few minor issues that should be addressed, however, with at least one major concern as follows.

We thank you for your positive and encouraging evaluation of our work. We also thank you for the insightful comments, which are helpful to improve our manuscript. We have sincerely revised the manuscript in line with the helpful suggestions, hoping that you find the revised version adequately improved.

Major concerns

1. Interesting data were provided using differential scanning calorimetry that Pib2 undergoes folding (increased heat capacity) upon binding L-glutamine. However, L-asparagine appears to produce a very strong effect (Fig 5C) that is even stronger when normalized by peak Cp's in the Trx domain. These data were described inaccurately in the Results as decreasing ΔH when it actually increased (Line 182). These findings raise the question of whether L-asparagine could bind Pib2 and interfere with the positive

effects of L-glutamine in the binding and activation assays of TORC1, which would suggest that Pib2 senses glutamine/asparagine ratios rather than glutamine itself. More experiments may be necessary to resolve this question.

We are afraid that our original description was misleading. The change we focused on here is about the sharp peak between 70°C and 80°C. As you rightfully pointed out, there appeared a noticeable broad peak between 40°C and 65°C in the case of L-Asn, and between 55°C and 65°C for others. We believe these broad peaks are from a loosely aggregated fraction of the purified protein because of the following reasons. Firstly, as we mentioned in our manuscript, recombinant Pib2 is highly susceptible to proteolytic degradation and aggregation (line 180-181). Secondly, the broad nature of the peaks suggests the heterogeneity of the underlying structures. Such irregular structures are prone to denature at relatively low temperatures and ligands at very high concentrations easily affect such non-specific structures. In fact, we have experienced that the size of these broad peaks in low temp ranges varied among different preparations. In recent experiments, improved technical fluency has remarkably reduced the broad peaks, and therefore we adopted data from new preparations for the new figure 5. In these experiments, the misleading broad peaks are less obvious.

On the other hand, we were curious about the possibility you have raised. To clarify if Pib2 senses the L-Gln/L-Asn ratios, we performed in vitro kinase assays (see data below). As you can see, we did not observe any L-Asn effects on the TORC1 activation. Therefore, we conclude that neither L-Asn nor the L-Gln/L-Asn ratio plays a role in the Pib2-mediated TORC1 activation.

2. Additionally, can the DSC experiment be replicated using instead the R341A variant of Pib2, which is reported here as critical for L-glutamine responsiveness? This simple experiment has the potential of resolving the issue above regarding L-asparagine as well as significantly increasing the resolution of the model.

Thank you for the insightful suggestion. According to your suggestion, we have performed the DSC experiment using the R341A variant. The results showed that R341A with L-Gln also exhibited higher ΔH (revised Fig.5d, lines 192-195). Therefore, we conclude that R341A retains the glutamine-sensing ability but loses the TORC1-binding ability. In fact, an isolated recombinant E motif polypeptide did not exhibit an L-Gln specific change by the DSC analysis (data not shown). Therefore, currently, we think that the E motif is responsible for the TORC1 interaction but not enough for (nor involved in) the L-Gln sensing itself.

Minor concerns

1. No clear orthologs of Pib2 have been reported in plants, and there is no data to suggest that the *Arabidopsis thaliana* protein included in the multiple sequence alignments (Fig 2C, 3A) interacts in any way with TORC1. Unless there is more data to support the inclusion of the plant sequences, they can be removed from the figures and the Discussion can be corrected without any loss of clarity.

According to the comment, we have removed the plant sequence from the multiple sequence alignments (Fig. 2c, 3a) and the Discussion.

2. Similarly, the E-domain of Pib2 is not conserved in *S. pombe* and by including this sequence (along with that of *A. thaliana*) in Fig 2C the color scheme used to represent conservation has been substantially altered. Both non-homologous sequences should be deleted from Fig 2C unless more evidence can be supplied regarding conservation.

According to the comment, we also eliminated the *S. pombe* sequence from Fig.2c and changed the color representing conservation.

Reviewer #3 (Remarks to the Author):

The manuscript by Tanigawa et al. investigates the role of Pib2 in the glutamine-dependent upstream regulation of TORC1 in budding yeast. The authors' aim is to gain more insights into the signal transduction of glutamine to TORC1 via direct binding to Pib2, which was previously shown in papers from their own group and others (Tanigawa and Maeda, 2017; Ukai et al., 2018). The characterization of the Pib2-dependent activation of TORC1 by L-glutamine is indeed of great interest for the community working on TORC1 upstream regulation. The manuscript is well written and the conclusions seem overall sound. However, some general technical aspects and specific issues would need to be addressed to properly back the claims of the paper.

We thank you for your positive and encouraging evaluation of our work. We also thank you for the insightful comments, which are helpful to improve our manuscript. We have sincerely revised the manuscript in line with the helpful suggestions, hoping that you find the revised version adequately improved.

A general remark: standard deviation bars (and the relative statistics) are represented only in Fig. 4, while I feel that this would be needed elsewhere in the paper, specifically where some graphs represent quantifications of blots where differences are hard to see by eye (see the specific comments below, this is especially needed in Fig. 5B).

Thank you for the constructive comment. According to your comment, we triplicated the experiments and added the standard deviation bars in Fig. 2b, 2e, and 5b.

Specific comments:

- Lines 102-105. The authors show side-by-side pull-downs and in vitro kinase assays with the same glutamine concentrations (Fig. 1C-D), which make sense. However, Fig. 1C is basically the same as in a previous paper (Ukai et al., 2018, Fig. 5B). Therefore, I propose to mention it in the text (“In agreement with...”).

We have reflected this comment by incorporating the statement below into the revised manuscript (lines 106-107).

“Glutamine dose-responsive Pib2-TORC1 interaction has also been reported previously²⁰.”

- Lines 119-120. “Pib2(304-620), which lacks the C-terminal tail motif essential for TORC1 activation 17,19,20, still interacted with TORC1 in response to glutamine (Fig.

2B),...”. This claim seems not to be supported by the blot, since in the pull-down of this Pib2 fragment both FLAG-Tor1 bands (with and without glutamine) are less intense than the respective controls (GST mock pull-downs).

We agree that our original data were not clear enough to claim the glutamine-responsive TORC1-Pib2(304-620) interaction. Thus, we added quantified results of the interactions between TORC1 and the Pib2 deletion mutants (Fig.2b). Both the basal and glutamine-stimulated TORC1-Pib2(304-620) interactions were weaker than TORC1-Pib2(304-635) interactions. Nonetheless, the fold-induction mediated by glutamine showed no significant difference between Pib2(304-635) and Pib2(304-620), supporting our claim that the tail motif is dispensable for the glutamine-responsive Pib2-TORC1 interaction.

· Lines 176-179. “In this assay, we used bacterially-expressed thioredoxin-tagged Pib2(304-533, Δ356-384), which lacks the large IDRs and is therefore relatively resistant to degradation and aggregation but retains its glutamine-induced interaction with TORC1 (Fig. 5A,B)”. The claimed 6-fold induced interaction in the presence of 30 mM glutamine is absolutely not observable from the blot. As abovementioned, this is the most relevant case where SDs (and maybe another blot) are needed in order to back the claim.

As you pointed out, the signal of Kog1-myc did not increase 6-fold by glutamine, although the graph represents Kog1-myc/GST-Pib2 ratio, leading to the increase exceeding 6-fold. However, thanks to your rightful comment, we were aware that the control GST blot was so uneven that the results could be incorrectly quantified. Therefore, we repeated the entire experiment and replaced Fig.5b with the new clearer data.

REVIEWERS' COMMENTS:

Reviewer #1 (Remarks to the Author):

The Authors have addressed all my concerns & explained their position on points where we did not agree. I find the changes made to the manuscript satisfactory.

Reviewer #2 (Remarks to the Author):

The authors addressed all my concerns. The manuscript now provides new insights into the regulation of TORC1 by glutamine

Reviewer #3 (Remarks to the Author):

I thank the authors of the manuscript "A glutamine sensor that directly activates TORC1" for addressing the points I raised during the revision. Therefore, I don't have any further remark about the manuscript, as presented in this revised form.